# Seed: Bridging Sequence and Diffusion Models for Road Trajectory Generation

## ABSTRACT

Road trajectory generation creates synthetic yet realistic trajectories to tackle data collection costs and privacy concerns. Existing methods generate a trajectory either segment-by-segment using sequence models or holistically in one step using diffusion models. Sequence-based models have good regularity and consistency (i.e., resemble the input trajectories) but lack diversity, while diffusion-based models enhance diversity but sacrifice regularity and consistency. To combine the merits of existing methods, we propose *Seed*, by bridging *se*quenc*e* and *di*ffusion models for trajectory generation. In particular, Seed adopts a *conditional diffusion structure*, where a Transformer models the movement of each trajectory along the road segments, and conditioned on the Transformer's output, a diffusion model recovers the next road segment from random noise. The rationale is that the Transformer captures sequential movement patterns for regularity and consistency, while the diffusion model introduces diversity by recovering from noise. To train Seed, we adopt Node2vec to learn embeddings for the road segments to prepare model inputs, supervise learning using the task of trajectory reconstruction, and design a curriculum learning strategy to accelerate convergence. We compare Seed with 8 state-of-the-art trajectory generation methods on 3 datasets, and the results show that Seed improves the best-performing baseline by over 50%.

## KEYWORDS

Trajectory, Data Synthesis, Transformer, Diffusion Model

**ACM Reference Format:**

. 2018. Seed: Bridging Sequence and Diffusion Models for Road Trajectory Generation. In *Proceedings of Make sure to enter the correct conference title from your rights confirmation emai (Conference acronym 'XX)*. ACM, New York, NY, USA, 12 pages. https://doi.org/XXXXXXX.XXXXXXX

## 1 INTRODUCTION

With the popularization of GPS devices, the movement of vehicles and individuals can be easily recorded as trajectories, and trajectory data facilitates many important applications, such as urban traffic planning [13, 29], vehicle navigation [15, 16], and route recommendation [6, 19]. However, obtaining real-world trajectories presents several challenges, including high data collection costs [23, 42], privacy concerns [33, 41], and proprietary restrictions [2, 35]. Trajectory generation [9, 14, 22, 25, 37, 39, 40] emerges as a solution to these challenges by creating synthetic yet realistic trajectories

**Unpublished working draft. Not for distribution.**

Table 1: Comparing SOTA trajectory generation methods with our Seed. *Type* indicates a method adopts recurrent (R) or holistic (H) generation. Consistency is quantified by the divergence between real and synthetic trajectories. Regularity is the percentage of trajectories that maintain connectivity. Diversity measures the portion of unique trajectories. For each metric, the top-3 methods are marked in bold.

| Methods | Type | Metrics | | |
|---|---|---|---|---|
| | | Consistency ($\downarrow$) | Regularity ($\uparrow$) | Diversity ($\uparrow$) |
| **SeqGAN** [39] | R | **0.0063** | 0.2549 | 0.8661 |
| **SVAE** [12] | R | 0.1188 | **1.0000** | 0.0001 |
| **TrajVAE** [3] | R | 0.0070 | **0.9994** | 0.2469 |
| **MoveSim** [7] | R | 0.0194 | 0.9916 | 0.1528 |
| **TS-TrajGen** [14] | R | **0.0026** | 0.6012 | 0.8697 |
| **DiffTraj** [43] | H | 0.0128 | 0.0000 | **1.0000** |
| **Diff-RNTraj** [37] | H | 0.0420 | 0.0000 | **1.0000** |
| **Seed (Ours)** | R | **0.0018** | **1.0000** | **0.9929** |

based on a reference trajectory dataset. To benefit the downstream applications, the synthetic trajectories are expected to resemble the reference trajectories (i.e., *consistency*) [7, 14], obey trajectory movement patterns (i.e., passing connected road segments, called *regularity*) [9, 37], differ from each other (i.e., *diversity*) [34, 43].

Existing methods for trajectory generation can be classified into two categories based on their methodology, i.e., *recurrent* and *holistic*. Recurrent methods utilize sequence models, such as Long Short-Term Memory (LSTM) [4] and Transformer [32], to generate trajectories in an auto-regressive manner (i.e., a road segment at a time). For instance, SeqGAN [39] trains an LSTM and a Generative Adversarial Network (GAN) using the policy gradient algorithm. TrajVAE [3] learns trajectory representations and reconstructs trajectories using LSTM and Variational Autoencoder (VAE). TS-TrajGen [14] uses Transformer and two GANs to generate each trajectory from coarse to fine granularity. Holistic methods generate a complete trajectory in a single step. For example, TrajGAN [25] utilizes a GAN based on Convolution Neural Network (CNN) to generate a virtual trajectory image, which is then converted to trajectory. To harness the power of diffusion models [11, 18, 20, 27, 31], DiffTraj [43] and Diff-RNTraj [37] integrate diffusion models into U-Net [28] and WaveNet [18] for trajectory generation.

As shown in Table 1, recurrent methods excel in consistency and regularity. This is because sequence models are good at capturing the movement patterns of the reference trajectories along the road segments. However, their diversity is not high because identical trajectories may be sampled following such movement patterns. In contrast, the two holistic methods that utilize diffusion models have high diversity but low consistency and regularity. This is because diffusion models recover trajectories from random noises, and thus it is unlikely to generate identical trajectories in different

runs. However, the diffusion models cannot capture the sequential movement patterns of trajectories along the road segments.

To tackle the limitations of existing researches, we aim to design Seed as a trajectory generation method that achieves consistency, regularity, and diversity simultaneously. The idea is to jointly utilize sequence and diffusion models to enjoy their merits while avoiding their defects. Although the idea sounds natural, two technical challenges need to be resolved as follows.

❶ *How to combine a sequence model with a diffusion model?* Seed utilizes a conditional diffusion structure, which generates a trajectory segment-by-segment like the recurrent methods and conditions the diffusion model on the sequence model when predicting the next road segment based on the previous road segments. In particular, Seed uses Transformer as the sequence model to encode the previous road segments passed by a trajectory. Different from standard diffusion models that take only random noise as input, Seed's diffusion model also uses the Transformer's output when recovering the next road segment. The idea is that the Transformer's output can guide the diffusion model to follow the movement patterns of the trajectories (e.g., prefer road segments that are related to the previous ones) while still injecting randomness during recovery. Besides, to transform the discrete road trajectories into continuous representations for model inputs, we adopt Node2vec to learning embeddings for the road segments.

❷ *How to train the two models effectively?* We utilize the trajectory reconstruction task for model training, which comprises a next-segment prediction task and a denoising task. The next-segment prediction task uses the cross-entropy loss and encourages the Transformer and diffusion model to work together and accurately predict the next road segment of a trajectory. We incorporate a spatial bias to enforce that the diffusion model can only sample road segments that are adjacent to the current road segment. The denoising task employs a noise-level loss and a sample-level loss to train the diffusion model. The noise-level loss minimizes the discrepancy between the added and model estimated noises, while the sample-level loss reduces error between the original and recovered road segment embeddings. Moreover, we utilize curriculum learning to train Seed from easy to more challenging tasks, which accelerates convergence and enhances consistency.

We conduct extensive experiments to evaluate Seed, employing 3 datasets and comparing with 8 state-of-the-art trajectory generation methods. The results show that Seed significantly outperforms all baselines in terms of consistency and matches the best-performing baselines in regularity and diversity. In particular, the consistency improvements of Seed over the best-performing baseline are 95.38% in the best case, 50.55% on average, and 9.09% in the worst case. We also conduct ablation study for our model designs and check the utility of the generated trajectories for downstream tasks. The results suggest that our designs contribute to performance, and the generated trajectories are effective for downstream tasks.

To summarize, we make the following contributions:

- We observe that existing trajectory generation methods cannot achieve consistency, regularity, and diversity at the same time due to the inherent limitations of their model designs.

- We propose Seed, which jointly utilizes sequence and diffusion models via a conditional diffusion structure, to combine the merits of existing methods while avoiding their defects.
- We design an effective procedure to train Seed, which includes preparing quality input embeddings, learning via next segment prediction, and utilizing curriculum learning for acceleration.

## 2 RELATED WORK

Existing solutions to the trajectory generation problem can be classified into two categories, i.e., *recurrent* and *holistic*.

**Recurrent methods.** They typically use sequence models, such as LSTM and Transformer, as a generator to generate trajectories in an auto-regressive manner. In particular, SeqGAN [39] employs an LSTM as the generator and a CNN-based discriminator, and trains them within a GAN framework using the policy gradient algorithm. MoveSim [7] uses Transformer as the generator and incorporates the sequential transition regularities of trajectories as prior knowledge into SeqGAN. SVAE [12] and TrajVAE [3] utilize two LSTMs or GRUs as the encoder and decoder, and train them within a VAE framework. The encoder learns trajectory representations, and the decoder uses the representations to reconstruct trajectories. TSG [36] and TS-TrajGen [14] generate trajectories from coarse to fine granularity in two-stages and train within a GAN framework. They first identify the regions passed by the trajectories, which are then used as conditions to generate the final trajectories. Using LSTM and Transformer as generators, DP-TrajGAN [41] and PateGail [33] adopt differential privacy techniques and a federated learning framework to generate privacy-preserving trajectories within the GAN framework. MobilityGPT [9] uses Generative Pre-trained Transformer (GPT) to generate trajectories and adopts a Reinforcement Learning from Trajectory Feedback (RLTF) mechanism for fine-tuning. ActSTD [40] and Volunteer [22] use Temporal Point Processes (TPP) to capture the spatial-temporal dynamics. ActSTD adopts generative adversarial imitation learning (GAIL) and the policy gradient algorithm to train the model, while Volunteer designs a two-layer VAE to extract user and trajectory information.

**Holistic methods.** They generate one trajectory in a single step. TrajGAN [25] and TrajGen [2] utilize standard CNN as both the generator and discriminator, training them within a GAN framework. They first generate a virtual trajectory image, which is then converted into trajectory. To incorporate the sequential transition patterns of trajectories, TrajGen also adopts a Seq2Seq model to assign movement order to the generated trajectories. To harness the powerful capabilities of diffusion models [11, 18, 20, 27, 31], recent studies propose to employ diffusion model for trajectory generation. DiffTraj [43] integrates a diffusion model into the U-Net [28] and uses trajectory attributes as extra conditions to achieve conditional generation. ControlTraj [44] uses Masked Autoencoder (MAE) to pretrain a road segment information extractor and incorporates road segment information into the condition used in DiffTraj. Diff-RNTraj [37] employs a diffusion model into WaveNet [18] to simultaneously generate road and GPS trajectories.

Existing trajectory generation methods cannot achieve consistency, regularity, and diversity simultaneously. Recurrent methods generate trajectories with high consistency and regularity but low

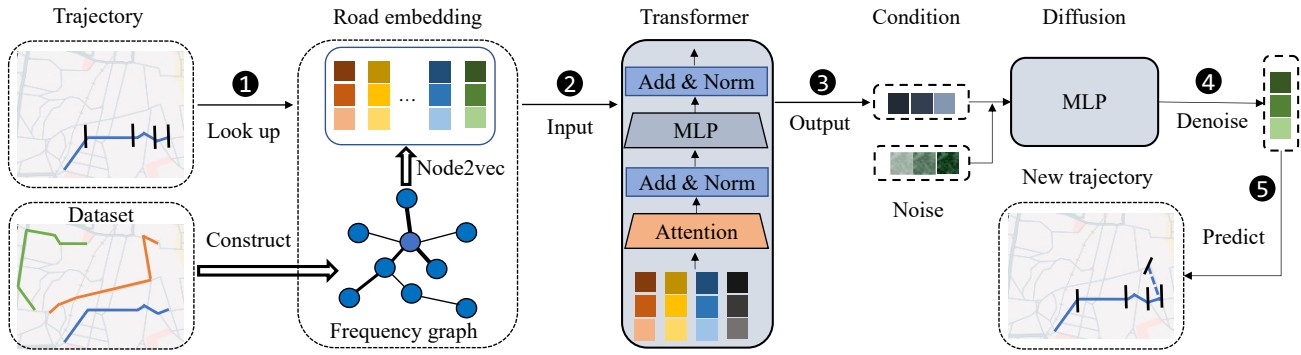

**Figure 1: An overview of Seed, where the numbers indicate the key steps.**

diversity, while holistic methods achieve high diversity at the expense of consistency and regularity. In contrast, our Seed excels in all three aspects by jointly utilizing the diversification capability of diffusion models and the movement modeling capability of sequence models.

## 3 PROBLEM DEFINITION

Here, we introduce the trajectory generation problem and relevant preliminary concepts.

**Definition 1: (Road Network).** A road network is a directed unweighted graph $\mathcal{R} = (\mathcal{V}, \mathcal{E})$, where $\mathcal{V}$ is the set of $|\mathcal{V}| = N$ road segments, and $\mathcal{E}$ is the set of $|\mathcal{E}| = M$ intersections among the road segments. The spatial connectivity between the road segments can also be modeled by an adjacency matrix $A$, where $A_{i,j} = 1$ if and only if road segments $v_i$ and $v_j$ are connected. □

**Definition 2: (Road Trajectory).** A GPS trajectory $\mathcal{T}_{gps}$ is a sequence of temporally ordered points. A road trajectory $\mathcal{T} = \{v_i | i = 1, 2, \cdots, n\}$ is a sequence of road segments with length $n$, which is derived from $\mathcal{T}_{gps}$ using a map matching algorithm [38]. □

**Definition 3: (Trajectory Generation).** Given a real-world trajectory dataset $D$, trajectory generation aims to learn a generative model to produce a synthetic road trajectory dataset $D'$. □

The main performance metrics for road trajectory generation are consistency, regularity, and diversity. Consistency measures the distribution divergence between the synthetic and real trajectories, regularity measures the percentage of synthetic trajectories that maintain connectivity, and diversity measures the portion of unique synthetic trajectories. For consistency, we use five metrics, i.e., Radius, G-rank, Density, Flow, and Location, following previous researches [7, 14, 37], with lower values indicating better consistency. Regularity is measured by two connectivity metrics, i.e., full connectivity (FC) and partial connectivity (PC), where higher values indicate better regularity. Diversity is quantified by one metric, i.e., the percentage of unique trajectories (UN), with higher values indicating greater diversity. More details about the performance metrics are provided in Appendix D.

## 4 SEED

In this part, we present our trajectory generation model Seed. As shown in Figure 1, Seed comprises three main components: i) a road

segment embedding module that provides a high-quality road segment embedding dictionary to convert the discrete road trajectories into continuous representations, ii) a Transformer that captures the transition patterns and provides meaningful guidance, and iii) a conditional diffusion module that learns the data distribution and generates road trajectories in an auto-regressive manner.

### 4.1 Road Segment Embedding

**Motivation.** To leverage the power of diffusion models, discrete trajectories must be transformed into continuous representations. A simple method uses a random road embedding dictionary $E \in \mathbb{R}^{N \times d}$, where each road segment in a trajectory is retrieved from $E$. However, this approach overlooks the road network topology and user travel patterns . To overcome this, we propose a pre-training strategy to learn a more effective road embedding dictionary.

**Pre-training strategy.** To capture road network topology, we attempt to employ the Node2vec algorithm [8] on the road network $\mathcal{R}$ to learn the road segment embedding dictionary $E \in \mathbb{R}^{N \times d}$. However, $\mathcal{R}$ treats each pair of road segments $(v_i, v_j)$ as equally important (i.e., unweighted graph), ignoring user preferences for these transitions. To overcome this limitation, we construct a transition frequency graph $\mathcal{G} = (\mathcal{V}, \mathcal{E}_t)$, where $\mathcal{E}_t$ represents the set of edges that capture transition frequencies between road segments. In this way, our learned road segment embeddings based on $\mathcal{G}$ not only preserve the topology of the road network but also capture the transition regularities between road segments. An alternative to using the graph $\mathcal{G}$ is to employ graph neural networks (GNNs) [10, 17, 26] to reflect the road network's topology and refine the road segment embeddings. However, we find that this approach incurs significant computational overhead and negatively affects model performance, prompting us to discard it.

### 4.2 Conditional Diffusion Structure

In this section, we provide a detailed explanation of how the diffusion model is employed to learn the data distribution and generate road trajectories in an auto-regressive manner.

**Preliminary on diffusion model.** Unlike standard diffusion model that operates on entire trajectory representations, we apply it to individual road segment embedding in an auto-regressive manner,

enhancing diversity during the generation process. The diffusion model mainly consists of the forward diffusion process and the reverse denoising process. The forward process adds Gaussian noise $\mathcal{N}(\cdot)$ to a road segment embedding $v_0$ over $T$ step: $q(v_{1:T}|v_0) = \prod_{t=1}^{T} q(v_t|v_{t-1}); q(v_t|v_{t-1}) = \mathcal{N}(v_t; \sqrt{1-\beta_t}v_{t-1}, \beta_t I)$. Here, $\{\beta_t \in (0,1)\}_{t=1}^{T} (\beta_1 < \beta_2 < \ldots < \beta_T)$ represents the variance schedule that controls the level of noise added at each forward step, and $I$ denotes the identity matrix. We can simplify the above equations to derive the distribution of $v_t$ conditioned on $v_0$ for each step as:

$$q(v_t|v_{t-1}) = \mathcal{N}(v_t; \sqrt{\bar{\alpha}_t}v_0; (1-\bar{\alpha}_t)I). \tag{1}$$

We adopt a re-parameterization trick to ensure the gradient remains differentiable [11]. Consequently, $v_t$ can be expressed as:

$$v_t = \sqrt{\bar{\alpha}_t}v_0 + \sqrt{1-\bar{\alpha}_t}\epsilon, \tag{2}$$

where $\epsilon \sim \mathcal{N}(0, I)$ and $\bar{\alpha}_t = \prod_{i=1}^{t}(1-\beta_i)$. The reverse process aims to recover the original road segment embedding from the noisy data $v_T$, which is formulated as: $p_\theta(v_{0:T}) = p(v_T)\prod_{t=1}^{T}p_\theta(v_{t-1}|v_t); p_\theta(v_{t-1}|v_t) = \mathcal{N}(v_{t-1}; \mu_\theta(v_t, t), \sigma_\theta(v_t, t)^2 I)$. Here, $\mu_\theta(v_t, t)$ and $\sigma_\theta(v_t, t)$ are the mean and variance predicted by a neural network with parameters $\theta$. We re-parameterize $\mu_\theta(v_t, t)$ as follows:

$$\mu_\theta(v_t, t) = \frac{1}{\sqrt{\alpha_t}}(v_t - \frac{\beta_t}{\sqrt{1-\bar{\alpha}_t}}\epsilon_\theta(v_t, t)), \tag{3}$$

where $\alpha_t = 1 - \beta_t$ and $\epsilon_\theta(v_t, t)$ represents the estimated noise at the current step $t$ based on the noised data $v_t$. The predicted variance is defined as $\sigma_\theta(v_t, t) = \frac{1-\bar{\alpha}_{t-1}}{1-\bar{\alpha}_t}\beta_t$.

**Guidance condition.** We use Transformer as sequence model to capture transition patterns, guiding the conditional diffusion model in denoising the corrupted next road segment embedding. Given a road trajectory $\mathcal{T} = \{v_j | j = 1, 2, \cdots, n\}$ and a road embedding dictionary $E$, we can obtain the continuous trajectory representation $X \in \mathbb{R}^{n \times d}$. The last element $v_n$ in $\mathcal{T}$ is removed and a start road segment $<CLS>$ is prepended, forming the inputs $\bar{X}$ to the Transformer. Formally, each road segment representation $v_i \in \bar{X}$ is feed into the attention layer to aggregate influences from relevant road segments: $z_i = \sum_{j=1}^{i}\alpha_{ij}(v_jW^V + p_j)$.

$$\alpha_{ij} = \frac{exp(\beta_{ij})}{\sum_{k=1}^{i}exp(\beta_{ik})}, \quad \beta_{ij} = \frac{v_iW^Q(v_jW^K + p_j)}{\sqrt{d}}, \tag{4}$$

where $W^Q, W^K, W^V \in \mathbb{R}^{d \times d}$ are weight matrices, and $p_j$ denotes positional embedding of $j$-th position. Here, we use relative positional encodings [30] because we find that relative positional encodings outperform absolute and learnable ones in our experiments. Subsequently, we employ feed-forward networks to capture non-linearity characteristics, which is computed as follows:

$$z_i = \phi(z_iW_1 + b_1)W_2 + b_2, \tag{5}$$

where $W_1, W_2 \in \mathbb{R}^{d \times d}$ are weight matrices, and $\phi$ is activation function. The remaining technical details, such as residual connections, layer normalization, and block stacking, are entirely in line with the principles of the Transformer. The combination of Transformer and the one-position offset introduced by the start road segment ensures that the road segment generated at the current position only

receives information from preceding road segments. This provides meaningful guidance for the auto-regressive generation.

**Road decoder.** As mentioned above, the output $z$ of Transformer at each step is treated as a condition, while the road decoder takes the corrupted target (next) road segment embedding $v_t$, produced by Equation 2, as input and predicts the clean one. Thus, the decoder effectively performs a denoising task and can be regarded as a conditional diffusion model. Formally, the reverse denoising process from the Equation 3 can be reformulated as:

$$\mu_\theta(v_t, t|z) = \frac{1}{\sqrt{\alpha_t}}(v_t - \frac{\beta_t}{\sqrt{1-\bar{\alpha}_t}}\epsilon_\theta(v_t, t|z)),$$
$$\sigma_\theta(v_t, t|z) = \sigma_\theta(v_t, t). \tag{6}$$

Here, we implement $\epsilon_\theta(v_t, t|z)$ using a Multi-Layer Perceptron (MLP), with the concatenation of $z$ and the corrupted target $v_t$ serving as the input. Since $\epsilon_\theta(\cdot)$ is responsible for predicting the added noise, it is necessary to recover the original road segment embedding to generate the corresponding discrete road segment:

$$\hat{v}_0 = (v_t - \sqrt{1-\bar{\alpha}_t}\epsilon_\theta(v_t, t|z))/\sqrt{\bar{\alpha}_t} \tag{7}$$

In order to convert the sampled road segment embedding $\hat{v}_0$ into the discrete road trajectory, we use an MLP to project $\hat{v}_0$ into the road segment space. An alternative is to calculate the similarity between the road segment dictionary $E$ and $\hat{v}_0$, and then select the road segment with the highest similarity. However, this method is time-consuming and yields worse performance in our experiments, so we have discarded it. To ensure the regularity in movement within a generated road trajectory, we refine the output of the projection MLP by a spatial bias. Formally, for the generation of $i$-th road segment, we have

$$\hat{v}_i = \arg\max_{v \in \mathbb{V}} \left(\text{sigmoid}(\text{MLP}(\hat{v}_0)) + A_{i-1,:}\right), \tag{8}$$

where $\text{sigmoid}(\cdot)$ ensures that the probability is non-negative, and $A_{i-1,:}$ reflects the connectivity of the $i-1$ road segment between others. In this way, we ensure that only the connections between actually connected roads are considered. Although other auto-regressive models, such as TrajVAE [3] and MoveSim [7], can also use this technique, they lead to deterministic outputs when given the same partial trajectory, thereby generating identical trajectories. Note that one row and one column, filled with 1 and 0s, are added to the matrix $A$ to accommodate the start road segment $<CLS>$.

## 4.3 Model Training

In this section, we use the trajectory reconstruction task for training and employ curriculum learning to accelerate model convergence and enhance model performance.

**Loss functions.** During the training phase, we adopt the cross-entropy function as our loss function for road segment prediction task, which is computed as follows:

$$\mathcal{L}_{CE} = -\sum_{i=1}^{n}\sum_{j=1}^{N}v_{i,j}\log\hat{v}_{i,j}, \tag{9}$$

where $v_{i,j}$ equals to 1 if $v_j$ is ground-truth road segment at the $i$-th time step and 0 otherwise. Additionally, DDPM [11] employs a

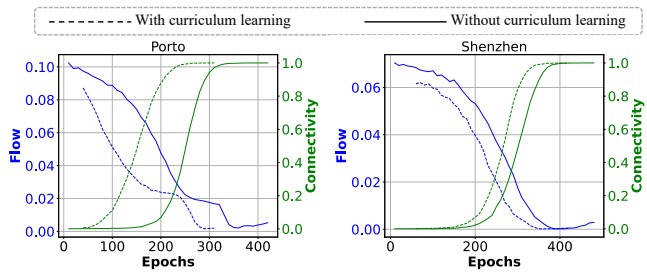

**Figure 2: The effect of curriculum learning. Lower is better for Flow, and higher is better for connectivity.**

noise-level loss function to minimize the error between the added noise $\epsilon$ and the estimated noise, which is defined as:

$$\mathcal{L}_{NL} = \mathbb{E}_{t,v_0,\epsilon}||\epsilon - \epsilon_\theta(\sqrt{\bar{\alpha}_t}v_0 + \sqrt{1-\bar{\alpha}_t}\epsilon, t|z)||^2. \quad (10)$$

To further incorporate the structure information of samples, we use a sample-level loss function to minimize the error between the original and the recovered sample representations:

$$\mathcal{L}_{SL} = ||v_0 - \hat{v}_0||^2. \quad (11)$$

Combined with the noise-level loss and sample-level loss, our multi-task learning objective is defined as follows:

$$\mathcal{L} = \mathcal{L}_{CE} + \mathcal{L}_{NL} + \lambda\mathcal{L}_{SL}, \quad (12)$$

where $\lambda = 1 \times 10^{-3}$ is used to balance the magnitudes of the three losses, due to the large scale of $\mathcal{L}_{SL}$. The detailed training algorithm is outlined in Algorithm 1 of Appendix.

**Curriculum learning.** As mentioned previously, given a road trajectory $\mathcal{T}$ and a road embedding dictionary $E$, the forward diffusion process randomly selects a noise level $t \in [0, T]$ for each road segment embedding $v$, producing noisy data $v_t$ according to Equation 2. However, when the noise level $t$ is large, the data $v_t$ becomes excessively noisy, obscuring useful information and making model training more difficult, slowing convergence, and negatively impacting model performance. To address this issue, we adopt a curriculum learning paradigm [1], which suggests that in the early stages of training, models are more sensitive to noise and difficult samples, and thus can benefit from a staged training approach. Specifically, in the first epoch, we set the noise level $t$ to 1, and then for the subsequent $K - 1$ epochs, we gradually increase the noise level each epoch according to:

$$t = \min(k * c, T),$$

where $k = 1, 2, \cdots, K - 1$ and $c$ is an adjustable parameter that controls the rate of difficulty increase. This approach accelerates model convergence by allowing the model to first learn the basic sequential patterns with lower noise. More importantly, it enhances model performance by providing a strong foundation during the early stages of training. To provide an intuitive illustration, we demonstrate the effectiveness and efficiency of curriculum learning on the Porto and Shenzhen datasets in Figure 2. As we can see, the dotted lines, representing the model with curriculum learning, achieve full connectivity of the generated road trajectories more quickly than the solid lines, which represent the model without curriculum learning. Additionally, the model with curriculum learning

exhibits better accuracy in consistency with the original dataset under the flow metric.

## 4.4 Model Inference

**Sampling.** The generation process can be summarized as follows: begin by initializing a partially generated trajectory with $< CLS >$. Then, sample the noisy data $v_T \sim \mathcal{N}(v_T; 0, I)$ and iteratively sample based on $v_{t-1} \sim p_\theta(v_{t-1}|v_t, z)$ to obtain the next road segment $v$, which is appended to the partial trajectory. This process continues in an auto-regressive manner until the final trajectory is generated. The algorithm is outlined in Algorithm 2 of Appendix.

**Sampling speed up.** Although our denoising network $\epsilon(\cdot)$ is more efficient than methods using more complex architectures, , such as DiffTraj [43] with UNet [28], Diff-RNTraj [37] with WaveNet [18], the auto-regressive generation process requires $O(nT)$ denoising steps to obtain the discrete road trajectories, which remains time-consuming. To address this issue, we adopt a non-Markov diffusion process following [31, 43], enabling more efficient reverse computations. Specifically, we sample every $S$ steps (e.g., $S = 5$) using the skip-step method from [24], which significantly reduces the number of sampling steps from $T$ to $\lceil T/S \rceil$ during road trajectory generation. Compared to standard diffusion models [11], this approach can generate samples with fewer steps.

## 5 EXPERIMENTAL EVALUATION

In this section, we present our experimental results on three real-world datasets that demonstrate the effectiveness of our approach. We experiment to answer the following research questions:

- *RQ1: How do the trajectory generation quality of Seed compare with those of state-of-the-art algorithms?*
- *RQ2: How do Seed's key components contribute to its performance?*
- *RQ3: How do the hyperparameters affect the performance of Seed?*
- *RQ4: How efficient are the training and inference processes of Seed?*
- *RQ5: How do the performances of Seed on downstream tasks compare with those of state-of-the-art algorithms?*

## 5.1 Experiment Settings

**Datasets.** We evaluate our Seed on three widely used real-world datasets: Porto[1], Shenzhen[2], and Chengdu[3]. We download the road networks from OpenStreetMap [5] and then apply the map matching algorithm [38] to convert GPS trajectories into road trajectories. For each dataset, we randomly select 80% of the trajectories for training and the remaining 20% for testing. The data statistics and more details are provided in Appendix A.

**Baselines.** We consider the following 8 representative methods as the baselines, including traditional method (Node2vec) that generates a random walk from a source road segment according to the probability, recurrent methods (SeqGAN, SVAE, TrajVAE, MoveSim, and TS-TrajGen) that adopt classical GAN and VAE frameworks

---

[1]https://www.kaggle.com/c/pkdd-15-predict-taxi-service-trajectory-i/data
[2]https://www.cs.rutgers.edu/~dz220/data.html
[3]https://js.dclab.run/v2/cmptDetail.html?id=175

Table 2: Performance of Seed and the baselines on the experiment datasets. In each column, the best and second-best methods are marked with boldface and underline, respectively. We use ↓ to indicate lower is better and ↑ to indicate the opposite.

| Datasets | Methods | Consistency | | | | | Regularity | | Diversity |
|---|---|---|---|---|---|---|---|---|---|
| | | Radius (↓) | Location (↓) | Density (↓) | Flow (↓) | G-rank (↓) | FC (↑) | PC (↑) | UN (↑) |
| **Porto** | Node2vec | 0.0115 | 0.0840 | 0.0266 | 0.1236 | 0.4731 | 0.3202 | 0.8782 | 0.0559 |
| | SeqGAN | 0.1045 | 0.0131 | 0.0063 | 0.0273 | 0.6175 | 0.2549 | 0.8759 | 0.8661 |
| | SVAE | 0.0115 | 0.1079 | 0.1188 | 0.1235 | 0.6931 | **1.0000** | **1.0000** | 0.0001 |
| | TrajVAE | 0.0024 | 0.0136 | 0.0070 | 0.0571 | 0.4590 | 0.9994 | **1.0000** | 0.2469 |
| | MoveSim | 0.0062 | 0.0299 | 0.0194 | 0.0725 | 0.6620 | 0.9916 | 0.9996 | 0.1528 |
| | TS-TrajGen | 0.1011 | 0.0011 | 0.0026 | 0.0248 | 0.0796 | 0.6012 | 0.8890 | 0.8697 |
| | DiffTraj | 0.6885 | 0.0183 | 0.0128 | 0.0125 | 0.2204 | 0.0000 | 0.0020 | **1.0000** |
| | Diff-RNTraj | 0.6868 | 0.1215 | 0.0420 | 0.0878 | 0.6931 | 0.0000 | 0.0003 | **1.0000** |
| | **Seed (Ours)** | **0.0002** | **0.0010** | **0.0018** | **0.0020** | **0.0293** | **1.0000** | **1.0000** | 0.9929 |
| | Improvement (%) | 91.67 | 9.09 | 30.77 | 84.00 | 63.19 | | - | |
| **Shenzhen** | Node2vec | 0.0332 | 0.0592 | 0.0222 | 0.0811 | 0.6329 | 0.4316 | 0.9150 | 0.0408 |
| | SeqGAN | 0.1858 | 0.0053 | 0.0004 | 0.0091 | 0.4568 | 0.2497 | 0.8732 | 0.9499 |
| | SVAE | 0.0341 | 0.0619 | 0.0239 | 0.0834 | 0.6931 | **1.0000** | **1.0000** | 0.0001 |
| | TrajVAE | 0.0017 | 0.0118 | 0.0012 | 0.0432 | 0.6774 | 0.9998 | **1.0000** | 0.5542 |
| | MoveSim | 0.0084 | 0.0085 | 0.0006 | 0.0665 | 0.6755 | 0.9938 | 0.9997 | 0.5527 |
| | TS-TrajGen | 0.0620 | 0.0006 | 0.0003 | 0.0018 | 0.0077 | 0.7516 | 0.9427 | 0.8679 |
| | DiffTraj | 0.6919 | 0.0007 | 0.0018 | 0.0019 | 0.0391 | 0.0000 | 0.0002 | **1.0000** |
| | Diff-RNTraj | 0.6902 | 0.0146 | 0.0035 | 0.0032 | 0.5789 | 0.0000 | 0.0002 | **1.0000** |
| | **Seed (Ours)** | **0.0004** | **0.0005** | **0.0002** | **0.0004** | **0.0030** | **1.0000** | **1.0000** | 0.9826 |
| | Improvement (%) | 76.47 | 16.67 | 33.33 | 77.78 | 61.04 | | - | |
| **Chengdu** | Node2vec | 0.0327 | 0.0444 | 0.0146 | 0.0525 | 0.3765 | 0.3541 | 0.7915 | 0.1400 |
| | SeqGAN | 0.0121 | 0.0028 | 0.0006 | 0.0035 | 0.5126 | 0.9139 | 0.9930 | 0.8958 |
| | SVAE | 0.0059 | 0.0522 | 0.0199 | 0.0558 | 0.6931 | **1.0000** | **1.0000** | 0.0000 |
| | TrajVAE | 0.0011 | 0.0159 | 0.0021 | 0.0324 | 0.6931 | 0.9992 | **1.0000** | 0.1971 |
| | MoveSim | 0.0046 | 0.0092 | 0.0009 | 0.0241 | 0.6931 | 0.9921 | 0.9996 | 0.1787 |
| | TS-TrajGen | 0.2495 | 0.0009 | 0.0003 | 0.0065 | 0.0303 | 0.4103 | 0.8118 | 0.8814 |
| | DiffTraj | 0.6767 | 0.0132 | 0.0020 | 0.0133 | 0.6931 | 0.0000 | 0.0070 | **1.0000** |
| | Diff-RNTraj | 0.6931 | 0.0111 | 0.0036 | 0.0069 | 0.5616 | 0.0000 | 0.0002 | **1.0000** |
| | **Seed (Ours)** | **0.0008** | **0.0005** | **0.0002** | **0.0003** | **0.0261** | **1.0000** | **1.0000** | 0.9977 |
| | Improvement (%) | 27.27 | 44.44 | 33.33 | 95.38 | 13.86 | | - | |

to train sequence models, and holistic methods (DiffTraj and Diff-RNTraj) that harness the powerful capabilities of diffusion models. The details of baselines are introduced in Appendix B.

## 5.2 Main Results (RQ1)

**Effectiveness study.** Table 2 shows the experiment results on Porto, Shenzhen, and Chengdu datasets. Based on these results, we have the following observations and corresponding analyses:

- The results from Porto, Shenzhen, and Chengdu datasets demonstrate that our proposed Seed significantly outperforms all other state-of-the-art baselines across all consistency evaluation metrics. Specifically, on the Porto dataset, Seed outperforms the second-best method with improvements of 91.76%, 9.09%, 30.77%, 84.00%, and 63.19% in Radius, Location, Density, Flow, and G-rank,

respectively. Additionally, Seed achieves average improvements of 53.06% and 42.86% compared to the second-best method on the Shenzhen and Chengdu datasets, respectively.

- Our Seed also achieves an excellent trade-off between regularity and diversity compared to other methods[4]. This can be attributed to several key factors: 1) By explicitly considering road connectivity and utilizing the Transformer's output as a condition for the diffusion model, Seed effectively captures important mobility patterns while ensuring diversity in the generated trajectories. 2) Both the pretrained road segment dictionary and the curriculum learning paradigm help mitigate training difficulties, thereby

---

[4]Since regularity and diversity should be considered together, it is unreasonable to calculate improvements for each metric separately; thus, we do not report individual metric improvements in Table 2.

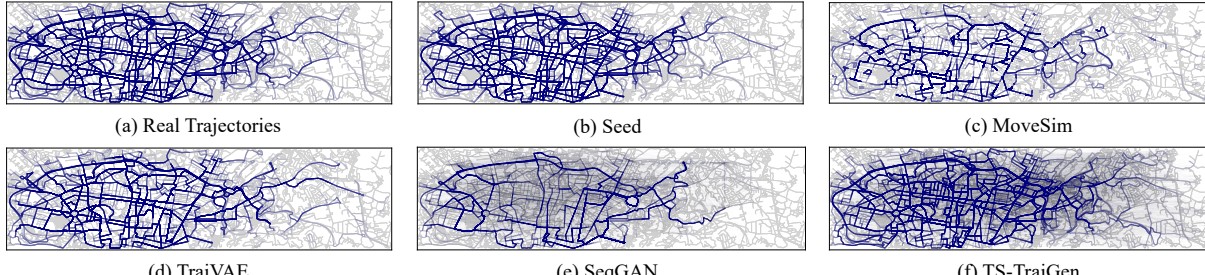

(a) Real Trajectories       (b) Seed       (c) MoveSim

(d) TrajVAE       (e) SeqGAN       (f) TS-TrajGen

**Figure 3: The trajectories generated by representative methods for the Porto dataset. Disconnected trajectories are plotted using more transparent lines, while connected trajectories are matched with the road network and plotted with opaque lines.**

enhancing model performance. Overall, these findings clearly demonstrate the superiority of our proposed Seed.

- Traditional method Node2vec maintains movement regularity but fails to capture real travel patterns, resulting in significant deviations from the original trajectories. This shows the challenge of representing complex human mobility with simple rules. Recurrent methods, such as SeqGAN and TS-TrajGen, demonstrate better regularity than holistic methods and outperform them across most consistency evaluation metrics. These advantages can be attributed to their ability to capture important mobility patterns between road segments in an auto-regressive manner. However, they exhibit lower diversity compared to holistic methods, as this generation manner may lead to deterministic outputs when given the same partial trajectory, resulting in the generation of identical trajectories.

**Geographic visualization.** To intuitively illustrate how our Seed outperforms the baselines, we depict the trajectory distributions generated by Seed and four representative baselines—SeqGAN, TrajVAE, MoveSim, and TS-TrajGen—on the Porto dataset. These baselines are selected for their ability to effectively capture regularity. Results from the holistic methods DiffTraj and Diff-RNTraj are omitted due to their poor regularity. All methods, including Seed, generate an equal number of road trajectories as the real test dataset. As shown in Figure 3, the visualizations indicate that all the generated trajectories accurately reflect the topology of the road network. Specifically, MoveSim and TrajVAE generate trajectories aligned with the road network, capturing regularity but lacking diversity, which limits their ability to fully reflect the original dataset. In contrast, SeqGAN and TS-TrajGen offer better diversity, capturing more dataset characteristics but suffer from lower regularity, leading to disconnected trajectories. Notably, Seed exhibits a clear representation of geographic density in the generated trajectories compared to the real test trajectories. Visualizations for the Shenzhen and Chengdu datasets are presented in Figure 7 and Figure 8.

## 5.3 Micro Results and Analysis

**Ablation study (RQ2).** There are three main components in our framework: (i) road embedding module, (ii) conditional diffusion model, and (iii) curriculum learning paradigm. To show the effects of these components, we conduct an ablation experiment on the Porto and Shenzhen datasets. To evaluate consistency, regularity,

**Table 3: Ablation study on Porto and Shenzhen. In each column, the best and second-best variants are marked with boldface and underline, respectively. We report only one metric for each aspect due to the space limit.**

| Variants | Porto | | | Shenzhen | | |
|---|---|---|---|---|---|---|
| | Location (↓) | FC (↑) | UN (↑) | Location (↓) | FC (↑) | UN (↑) |
| **Seed** | **0.0010** | **1.0000** | 0.9929 | **0.0004** | **1.0000** | 0.9826 |
| *w/o Diffusion* | 0.1079 | 1.0000 | 0.0001 | 0.0834 | 1.0000 | 0.0001 |
| *w/o Transformer* | 0.0183 | 0.0000 | **1.0000** | 0.0007 | 0.0000 | **1.0000** |
| *w/o Pretrain* | 0.0184 | 1.0000 | 0.8244 | 0.0083 | 1.0000 | 0.8179 |
| *w/o TGraph* | 0.0018 | 1.0000 | 0.9893 | 0.0014 | 1.0000 | 0.9702 |
| *w/o Curriculum* | 0.0038 | 1.0000 | 0.9848 | 0.0029 | 1.0000 | 0.9549 |
| *w/o SL* | 0.0265 | 1.0000 | 0.4521 | 0.0082 | 1.0000 | 0.7932 |

and diversity, we select one metric for each, with the results shown in Table 3 (full results are provided in Appendix E). The base model is Seed and we form the different variants as follows:

- *w/o Diffusion*: This variant removes the diffusion model and uses the output of Transformer to predict next road segment.
- *w/o Transformer*: This variant removes the Transformer and uses diffusion model to predict overall trajectory in a single step.
- *w/o Pretrain*: This variant uses an initially randomized road embedding that fails to capture the road network's topology.
- *w/o TGraph*: This variant replaces the transition graph with the road network to learn the road embedding dictionary.
- *w/o Curriculum*: This variant removes the curriculum learning.
- *w/o SL*: This variant removes the loss $\mathcal{L}_{SL}$, i.e., $\lambda = 0$.

Based on the statistics from Table 3, we have the following findings:

- We observe that the diffusion model and Transformer are the most critical components. Without *Diffusion*, the method degrades to vanilla autoregressive approaches like SVE, producing many identical trajectories and missing crucial mobility patterns. Without *Transformer*, it reverts to standard diffusion models like DiffTraj, failing to capture the regularity.
- We notice that both *w/o Pretrain* and *w/o TGraph* exhibit a significant performance drop compared to Seed, with the former performing worse than the latter. This indicates that utilizing the transition graph is beneficial for simultaneously capturing the topology structure and user travel patterns.

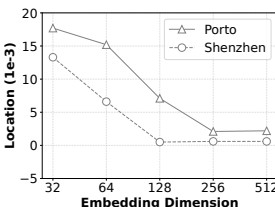

**(a) Embedding dimension $d$**

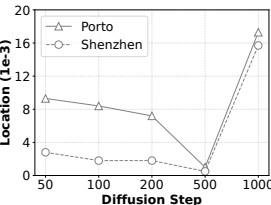

**(b) Diffusion steps $T$**

**Figure 4: Impact of Seed's hyper-parameters on consistency.**

- We find that the curriculum learning plays a crucial role in guiding the training of our Seed from easier to more challenging tasks, significantly accelerating model convergence speed and enhancing the overall model performance. Additionally, the sample-level loss $\mathcal{L}_{SL}$ helps to capture the structure information of samples, providing more supervision signals for effective training.

**Hyper-parameters (RQ3).** We conduct experiments to analyze the impacts of two critical hyper-parameters on the Porto and Shenzhen datasets: embedding dimension $d$, and diffusion steps $T$. The results are presented in Figure 4. Specifically, Figure 4a demonstrates that consistency improves as $d$ increases and then stabilizes. This indicates a large dimension may adequately represent the topology of the road network. Therefore, we choose $d = 256$ and $d = 128$ for the Porto and Shenzhen datasets. Figure 4b shows that consistency improves as $T$ increases; however, further increasing $T$ can negatively impact the curriculum learning process, leading to ineffective model training. The impacts of more hyper-parameters on the Porto and Shenzhen datasets are presented in Appendix F.

**Efficiency analysis (RQ4).** Table 4 compares the average training and test time of Seed against five representative methods on the Porto and Chengdu datasets, including GAN-based methods (SeqGAN, MoveSim, TS-TrajGen), a VAE-based method (TrajVAE), and a diffusion model-based method (DiffTraj). During training, we measure the average time per trajectory by dividing the total time by the number of trajectories. In the test phase, we compute the average time for generating a single trajectory. GAN-based methods require iterative generation of synthetic trajectories to train a discriminator, leading to higher time consumption compared to VAE-based method TrajVAE. MoveSim and TS-TrajGen take considerably more time than SeqGAN, as MoveSim employs GNN-like matrix multiplication operations, and TS-TrajGen uses the A* algorithm for road segment search. Although DiffTraj generates trajectories in a single step, its complex model design make it less efficient than TrajVAE. In contrast, while Seed generates trajectories step-by-step, its simple MLP-based diffusion model significantly reduces the time consumption. In summary, the results show that the training and test time of Seed is not long and ranks in the middle among the baselines.

**Case study: Next location prediction (RQ5).** The next location prediction task aims to forecast the subsequent location in a mobility trajectory by mining mobility patterns. We can leverage this task to assess whether real mobility patterns exist in the generated trajectories. Specifically, we use the gated recurrent unit (GRU) [4]

**Table 4: The average training and test time (in ms) of each trajectory on the Porto and Chengdu datasets.**

| Methods | Porto | | Chengdu | |
|---|---|---|---|---|
| | Train | Test | Train | Test |
| SeqGAN | 11.16 | 0.023 | 33.69 | 0.033 |
| TrajVAE | 0.135 | 0.023 | 0.313 | 0.033 |
| MoveSim | 38.71 | 1.031 | 82.73 | 3.395 |
| TS-TrajGen | 1117 | 317.7 | 4398 | 407.0 |
| DiffTraj | 0.223 | 1.018 | 0.282 | 1.013 |
| **Seed** | 0.165 | 0.739 | 0.363 | 1.926 |

**Table 5: Next location prediction accuracy using the synthetic trajectories produced by different methods. In each column, we use boldface and underline to indicate the best and second-best methods, respectively.**

| Methods | Porto | | Shenzhen | |
|---|---|---|---|---|
| | Hit@5 | Hit@10 | Hit@5 | Hit@10 |
| SeqGAN | 0.881 | 0.891 | 0.859 | 0.861 |
| TrajVAE | 0.008 | 0.010 | 0.004 | 0.008 |
| MoveSim | 0.585 | 0.587 | 0.605 | 0.607 |
| TS-TrajGen | 0.915 | 0.922 | 0.845 | 0.846 |
| DiffTraj | 0.001 | 0.002 | 0.000 | 0.001 |
| **Seed** | 0.926 | 0.927 | 0.895 | 0.895 |
| Real | **0.945** | **0.946** | **0.944** | **0.944** |

as the prediction model and employ Hit Ratio (HR) as the accuracy metric. We randomly sample an equal number of trajectories from the training set as those in the test dataset to train the model and evaluate its accuracy on the test dataset, which can be considered an upper bound. We then train the model using the generated trajectories from various baselines and evaluate its accuracy on the same test dataset. As shown in Table 5, the accuracy of our Seed is closer to that of the real training set, indicating that more mobility patterns are captured in the generated trajectories. Additionally, most baselines achieve higher accuracy on the Porto dataset compared to the Shenzhen dataset, due to the greater sparsity of the Shenzhen dataset. The another case study on trajectory outlier detection can be found in Appendix G.

## 6 CONCLUSION

In this paper, we propose a conditional diffusion model framework (Seed) for road trajectory generation task. We use Transformer to extract the movement pattern between road segments, which is regarded as the guidance condition of an MLP-based diffusion model. To train the two models effectively, we propose a trajectory reconstruction task and design a curriculum learning paradigm. Additionally, we propose a pre-training strategy to learn a high-quality road embedding dictionary. Experimental results on three real-world datasets show the superiority of our Seed.

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

**Table 6: Datasets Statistics**

| Dataset | # Traj | # Segments | Distance |
|---------|--------|------------|----------|
| Porto | 96,896 | 8,476 | 1,920m |
| Shenzhen | 42,011 | 19,718 | 2,863m |
| Chengdu | 334,232 | 22,332 | 2,820m |

[33] Huandong Wang, Changzheng Gao, Yuchen Wu, Depeng Jin, Lina Yao, and Yong Li. 2023. PateGail: a privacy-preserving mobility trajectory generator with imitation learning. In *AAAI*.
[34] Jiawei Wang, Renhe Jiang, Chuang Yang, Zengqing Wu, Makoto Onizuka, Ryosuke Shibasaki, and Chuan Xiao. 2024. Large Language Models as Urban Residents: An LLM Agent Framework for Personal Mobility Generation. *NeurIPS* (2024).
[35] Jinzhong Wang, Xiangjie Kong, Feng Xia, and Lijun Sun. 2019. Urban human mobility: Data-driven modeling and prediction. *ACM SIGKDD explorations newsletter* (2019).
[36] Xingrui Wang, Xinyu Liu, Ziteng Lu, and Hanfang Yang. 2021. Large scale GPS trajectory generation using map based on two stage GAN. *Journal of Data Science* (2021).
[37] Tonglong Wei, Youfang Lin, Shengnan Guo, Yan Lin, Yiheng Huang, Chenyang Xiang, Yuqing Bai, Menglu Ya, and Huaiyu Wan. 2024. Diff-rntraj: A structure-aware diffusion model for road network-constrained trajectory generation. *arXiv* (2024).
[38] Can Yang and Gyozo Gidofalvi. 2018. Fast map matching, an algorithm integrating hidden Markov model with precomputation. *International Journal of Geographical Information Science* (2018).
[39] Lantao Yu, Weinan Zhang, Jun Wang, and Yong Yu. 2017. Seqgan: Sequence generative adversarial nets with policy gradient. In *AAAI*.
[40] Yuan Yuan, Jingtao Ding, Huandong Wang, Depeng Jin, and Yong Li. 2022. Activity trajectory generation via modeling spatiotemporal dynamics. In *SIGKDD*.
[41] Jing Zhang, Qihan Huang, Yirui Huang, Qian Ding, and Pei-Wei Tsai. 2023. DP-TrajGAN: A privacy-aware trajectory generation model with differential privacy. *FGCS* (2023).
[42] Yu Zheng. 2015. Trajectory data mining: an overview. *ACM Transactions on Intelligent Systems and Technology (TIST)* (2015).
[43] Yuanshao Zhu, Yongchao Ye, Shiyao Zhang, Xiangyu Zhao, and James Yu. 2023. Difftraj: Generating gps trajectory with diffusion probabilistic model. *NeurIPS* (2023).
[44] Yuanshao Zhu, James Jian Qiao Yu, Xiangyu Zhao, Qidong Liu, Yongchao Ye, Wei Chen, Zijian Zhang, Xuetao Wei, and Yuxuan Liang. 2024. ControlTraj: Controllable Trajectory Generation with Topology-Constrained Diffusion Model. In *SIGKDD*.

## A DATASETS.

Table 6 shows the statistics of the three datasets used in our experiments, detailing the number of trajectories, road segments, and the average trajectory distance. Each dataset comprises original GPS trajectories, defined as a sequence of temporally ordered points, with each point including latitude, longitude, and timestamp, respectively. Specifically, the Porto dataset contains GPS trajectories of 442 taxis in Porto, Portugal, recorded between January 2013 and June 2014, with points sampled every 15 seconds. The Shenzhen dataset includes trajectories from 11,100 taxis in Shenzhen, China, with an average sampling rate of every 15 seconds. The Chengdu dataset contains over 1.4 billion trajectory points from approximately 14,000 taxis in Chengdu, China, collected from August 3 to August 30, 2014, with points sampled every 30 seconds on average. For our experiments, we use a two-day subset of the Chengdu dataset. Each dataset's road network is downloaded from OpenStreetMap [5], and we apply a map matching algorithm [38] to convert GPS trajectories into road trajectories. We divide each road trajectory of length $m$ into multiple sub-trajectories of length $n$, provided that $m \geq n$; otherwise, the trajectory is discarded. Additionally, we filter out duplicated trajectories. Finally, for each

dataset, we randomly select 80% of the trajectories for training and reserve the remaining 20% for testing.

## B  BASELINES

We provide the details of baseline methods in our experiments.

- **Node2vec** [8]: It randomly selects an origin, then performs the random walk on the road network according to the probability.
- **SeqGAN** [39]: It is a sequential generative model that trains trajectories in discrete sequences with policy gradient algorithm.
- **SVAE** [12] & **TrajVAE** [3]: They learn the trajectory representation using LSTM and VAE, and reconstruct the road trajectory through a LSTM-based decoder.
- **MoveSim** [7]: It is a state-of-the-art method that incorporates transition regularities as prior knowledge into the SeqGAN model.
- **TS-TrajGen** [14]: It is a state-of-the-art method that designs a two-stage training procedure and a modified $A^*$ search algorithm.
- **DiffTraj** [43]: It is a state-of-the-art method that employs a diffusion model into U-Net for continuous trajectory generation.
- **Diff-RNTraj** [37]: It is a state-of-the-art method that employs a diffusion model into WaveNet for discrete trajectory generation.

## C  IMPLEMENTATION DETAILS

We use the Adam optimizer with default betas and a learning rate of 0.0001. The embedding dimension $d$, the number of epochs $K$ for curriculum learning, and the difficulty level $c$ are set to 256, 50, and 3 for the Porto dataset; 128, 60, and 5 for the Shenzhen dataset; and 256, 3, and 5 for the Chengdu dataset. Additionally, we set the number of the diffusion steps T to 500, and adopt a linear schedule for the variance schedule $\beta$, with a minimum noise level of $\beta_1 = 0.0001$ and a maximum noise level of $\beta_T = 0.05$. Our code is available at https://anonymous.4open.science/r/Seed-AD52.

## D  EVALUATION METRICS

The main performance metrics for road trajectory generation are consistency, regularity, and diversity. For consistency, we use five metrics—Radius, G-rank, Density, Flow, and Location—following previous works [7, 14, 37], employing Jensen-Shannon divergence (JSD) to evaluate distribution divergence between $p$ and $q$:

$$\text{JSD}(p, q) = \frac{1}{2}\text{KL}(p||\frac{p+q}{2}) + \frac{1}{2}\text{KL}(q||\frac{p+q}{2}),$$

where $\text{KL}(\cdot||\cdot)$ denote the Kullback-Leibler (KL) divergence. The detailed descriptions of five metrics are as follows:

- **Radius**: The spatial range distribution, which is calculated as the root mean square distance of all points from the central one.
- **Location**: The distribution of road segment, which is calculated as the frequency of each road segment.
- **Density**: The distribution of grid, which is calculated as the frequency of each grid mapped by each road segment.
- **Flow**: The distribution of end road segment, which is calculated as the frequency of each end road segment.
- **G-rank**: The number of visits per road segment, which is calculated as the visiting frequency of top-100 road segments.

For regularity, we use two connectivity metrics to quantify the percentage of trajectories that maintain regularity in movement:

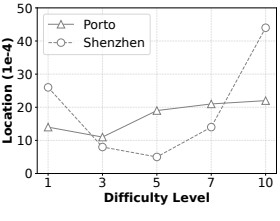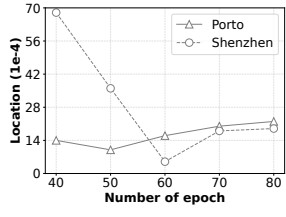

(a) difficulty level $c$      (b) number of epochs $K$

**Figure 5: Impact of Seed's hyper-parameters on consistency.**

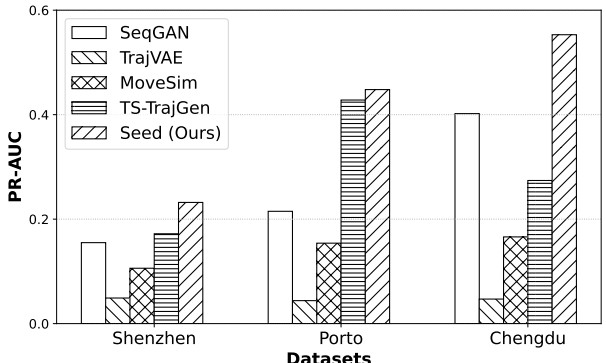

**Figure 6: Trajectory outlier detection on all three datasets.**

- **FC**: FC quantifies the percentage of the generated trajectories that are fully connected from origin to destination.
- **PC**: PC measures the percentage of adjacent road segments within generated trajectories are reachable in the road network.

  For diversity, we use one metric to quantify the percentage of trajectories that are unique in the generated trajectories $\mathcal{T}_g$:

- **UN**: UN quantifies the portion of the unique trajectories, which is calculated as $UN = |\text{Unique}(\mathcal{T}_g)|/|\mathcal{T}_g|$.

## E  ABLATION STUDY

We conduct an ablation experiment on the Porto, Shenzhen, and Chengdu datasets. The results are shown in Table 7.

## F  PARAMETER STUDY

We conduct experiments to analyze the impacts of another two critical hyper-parameters on the Porto and Shenzhen datasets: difficulty level $c$, and number of epochs $K$. As shown in Figure 5, the consistency first decreases then increases. This behavior arises because both parameters influence the curriculum learning process. Too few epochs or a low difficulty level cause the model to primarily encounter easy samples, while too many epochs or a high difficulty level expose it to many difficult samples, both of which hinder effective training.

## G  TRAJECTORY OUTLIER DETECTION.

Trajectory outlier detection is another crucial task in trajectory mining, aiming to identify trajectories that significantly deviate

Table 7: Overall ablation study on three experiment datasets.

| Datasets | Methods | Consistency | | | | | Regularity | | Diversity |
|---|---|---|---|---|---|---|---|---|---|
| | | Radius ($\downarrow$) | Location ($\downarrow$) | Density ($\downarrow$) | Flow ($\downarrow$) | G-rank ($\downarrow$) | FC ($\uparrow$) | PC ($\uparrow$) | UN ($\uparrow$) |
| Porto | **Seed** | **0.0002** | **0.0010** | **0.0018** | **0.0020** | **0.0293** | **1.0000** | **1.0000** | 0.9929 |
| | w/o Diffusion | 0.0115 | 0.1079 | 0.1185 | 0.1235 | 0.6931 | 1.0000 | 1.0000 | 0.0001 |
| | w/o Transformer | 0.6885 | 0.0183 | 0.0128 | 0.0125 | 0.2204 | 0.0000 | 0.0020 | **1.0000** |
| | w/o Pretrain | 0.0007 | 0.0184 | 0.0161 | 0.0156 | 0.6013 | 1.0000 | 1.0000 | 0.8087 |
| | w/o TGraph | 0.0003 | 0.0018 | 0.0021 | 0.0030 | 0.0446 | 1.0000 | 1.0000 | 0.9893 |
| | w/o Curriculum | 0.0003 | 0.0038 | 0.0048 | 0.0053 | 0.2964 | 1.0000 | 1.0000 | 0.9848 |
| | w/o SL | 0.0028 | 0.0265 | 0.0241 | 0.0277 | 0.6628 | 1.0000 | 1.0000 | 0.4521 |
| Shenzhen | **Seed** | **0.0004** | **0.0005** | **0.0002** | **0.0004** | **0.0030** | **1.0000** | **1.0000** | 0.9826 |
| | w/o Diffusion | 0.0341 | 0.0619 | 0.0240 | 0.0834 | 0.6931 | 1.0000 | 1.0000 | 0.0001 |
| | w/o Trans | 0.6919 | 0.0007 | 0.0018 | 0.0019 | 0.0391 | 0.0000 | 0.0002 | **1.0000** |
| | w/o Pretrain | 0.0032 | 0.0110 | 0.0029 | 0.0083 | 0.6931 | 1.0000 | 1.0000 | 0.8179 |
| | w/o TGraph | 0.0006 | 0.0013 | 0.0004 | 0.0014 | 0.0610 | 1.0000 | 1.0000 | 0.9702 |
| | w/o Curriculum | 0.0009 | 0.0029 | 0.0005 | 0.0029 | 0.2789 | 1.0000 | 1.0000 | 0.9549 |
| | w/o SL | 0.0013 | 0.0082 | 0.0018 | 0.0044 | 0.6700 | 1.0000 | 1.0000 | 0.7932 |
| Chengdu | **Seed** | **0.0008** | **0.0005** | **0.0002** | **0.0003** | **0.0261** | **1.0000** | **1.0000** | 0.9977 |
| | w/o Diffusion | 0.0059 | 0.0524 | 0.0201 | 0.0558 | 0.6931 | 1.0000 | 1.0000 | 0.0000 |
| | w/o Transformer | 0.6767 | 0.0132 | 0.0020 | 0.0133 | 0.6931 | 0.0000 | 0.0070 | **1.0000** |
| | w/o Pretrain | 0.0008 | 0.0069 | 0.0013 | 0.0055 | 0.6691 | 1.0000 | 1.0000 | 0.9869 |
| | w/o TGraph | 0.0008 | 0.0007 | 0.0004 | 0.0007 | 0.1266 | 1.0000 | 1.0000 | 0.9967 |
| | w/o Curriculum | 0.0008 | 0.0042 | 0.0009 | 0.0037 | 0.4409 | 1.0000 | 1.0000 | 0.9907 |
| | w/o SL | 0.0010 | 0.0045 | 0.0012 | 0.0028 | 0.4996 | 1.0000 | 1.0000 | 0.9867 |

---

**Algorithm 1:** The training processes of Seed

**Input:** Trajectories dataset $\mathcal{T}_D = \{\mathcal{T}_1, \ldots, \mathcal{T}_{|D|}\}$; Forward diffusion step $T$; Pretrained road segment dictionary $E$; Variable schedule $\{\beta_1, \ldots, \beta_T\}$.

**Output:** Model $\Theta$.

1 **while** *model not converged* **do**
2     Sample a set of real data samples $\mathcal{X}$ from $\mathcal{T}_D$;
3     Retrieve $E$ to obtain the representations $X$;
4     Sample $t \sim [0, T]$, and $\epsilon \sim \mathcal{N}(0, I)$;
5     Obtain noisy data $v_t$ based on Equation 2;
6     Calculate the condition $z$ using Equation 4;
7     Calculate the estimated noise $\epsilon_\theta(v_t, t|z)$;
8     Calculate the recovered representations $\hat{v}_0$;
9     Calculate the loss $\mathcal{L}$ according to Equation 12;
10     Back-propagate and update parameters;
11 Return Model;

---

**Algorithm 2:** The inference processes of Seed

**Input:** Forward diffusion step $T$; Pretrained road segment dictionary $E$; Trained model $\Theta$; the length of trajectory $n$; Variable schedule $\{\beta_1, \ldots, \beta_T\}$.

**Output:** A generated road trajectory.

1 Initialize road trajectory $\mathcal{T} \leftarrow [<CLS>]$;
2 **for** $i \leftarrow 1$ **to** $n$ **do**      // trajectory length
3     Sample noisy data $v_T \sim \mathcal{N}(v_T; 0, I)$;
4     Calculate the condition $z$ using Equation 4;
5     **for** $t = T, T-S, \ldots, 1$ **do**    // Sampling speed up
6         Calculate the estimated noise $\epsilon_\theta(v_t, t|z)$;
7         Sample data $v_{t-1} \sim p_\theta(v_{t-1}|v_t, z)$;
8     Calculate the recovered representations $\hat{v}_0$;
9     Obtain the road segment $v_i$ based on Equation 8;
10     Append $v_i$ into the road trajectory $\mathcal{T}$;
11 Return the road trajectory $\mathcal{T}$;

---

from normal patterns. This task requires high-quality trajectories to accurately capture the characteristics of normal trajectories and effectively distinguish anomalous ones. Specifically, we use GM-VSAE [21] as the detection model, randomly generate 5% anomalous trajectories, and use the PR-AUC as the evaluation metric. To construct anomalous trajectories, we use a parameter $\alpha$ to control the proportion of road segments that are required to change their order in the original trajectories. For example, $\alpha = 0.1$ means that 10% of the road segments in a trajectory will have their order changed. In our experiments, we set $\alpha$ to 0.2 and follow the official implementation of GM-VSAE [21]. The road segment embedding dimension is set to 128, while the hidden sizes of the encoder and decoder are both set to 512. The number of Gaussian components is set to 10. As shown in Figure 6, the model trained on trajectories generated by Seed achieves best performance, indicating that the characteristics of normal trajectories are effectively captured.

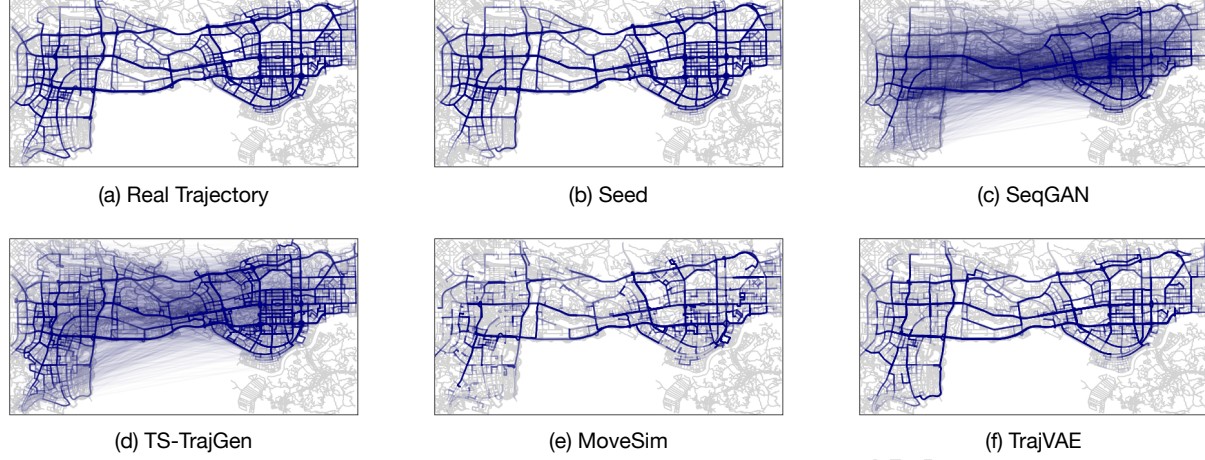

Figure 7: Visualization of the different methods on Shenzhen dataset. The disconnected trajectories are shown with more transparent lines, while the connected trajectories are matched with the road network and displayed with opaque lines.

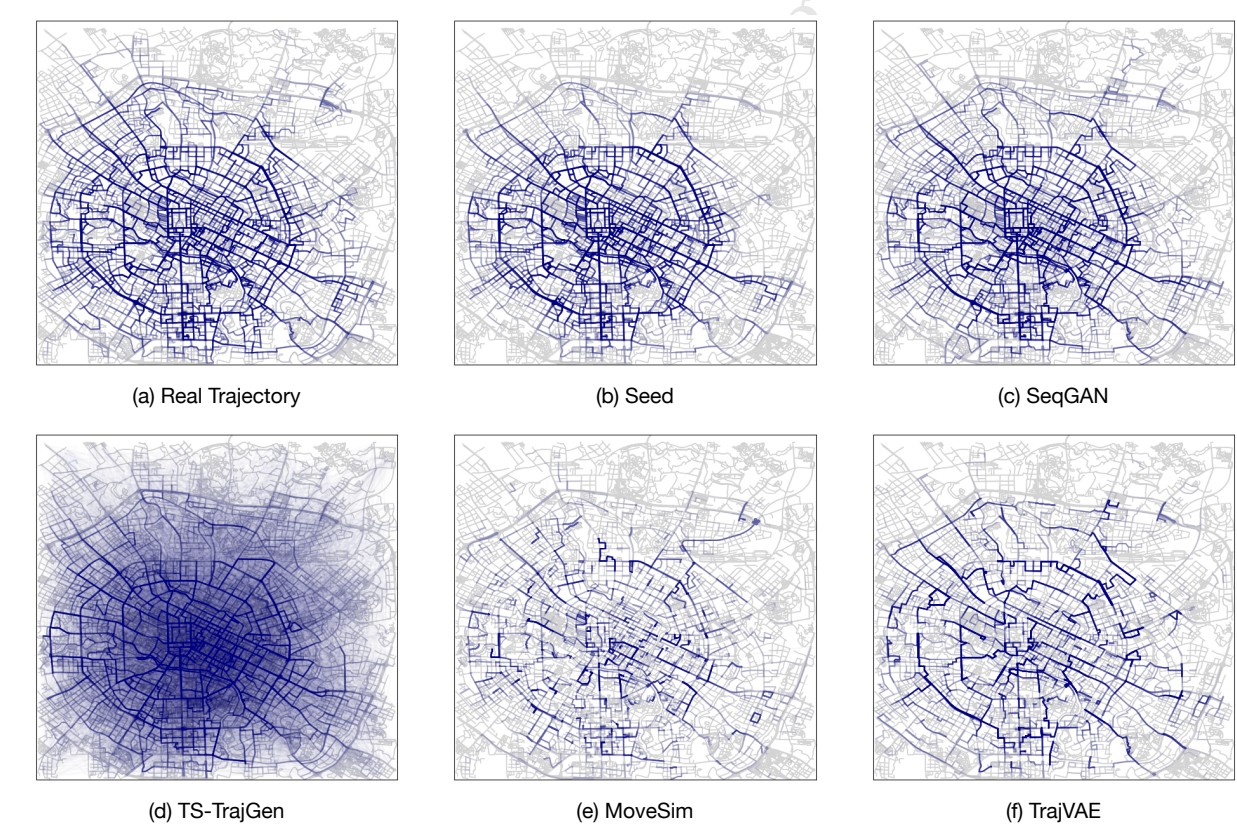

Figure 8: Visualization of the different methods on Chengdu dataset. The disconnected trajectories are shown with more transparent lines, while the connected trajectories are matched with the road network and displayed with opaque lines.

