# OpenReview forum: "Seed: Bridging Sequence and Diffusion Models for Road Trajectory Generation"
_ACM.org/TheWebConf/2025/Conference — WWW 2025 Poster_

### Official Review · Reviewer_gLNk · 2024-11-13

**Novelty:** 6
**Technical Quality:** 6

**Review:**

The work incorporates sequence-based models and diffusion-based models to build a conditional diffusion structure (Seed). Seed generates synthetic road trajectories. The generated trajectories feature consistency and regularity (from sequence models) and diversity (from diffusion models). Seed’s model is trained through trajectory reconstruction, with accelerated training via curriculum learning.


The motivation and contribution are significant, the method is sound, clear, and original, and the evaluation is comprehensive. The proposed solution exhibits a superior performance, validated through several valid experiments and performance metrics.

**Pros:**
- The motivation is strong and the research problem is intriguing.
- The paper is well-written, presented, and structured.
- The performance metrics selected are suitable to prove the efficiency of the proposed solution.
- The evaluation is extensive and comprehensive and the proposed framework (Seed) shows a promising performance for generating synthetic trajectories that are diverse and passing connected road segments.

**Cons:**
-  This work does not seem very relevant to the Web

**Questions:**

The paper doesn’t go into why this work is important for the Web, so the authors should clarify its relevance.

**Reviewer Confidence:**

2: The reviewer is willing to defend the evaluation, but it is likely that the reviewer did not understand parts of the paper

**Scope:**

2: The connection to the Web is incidental, e.g., use of Web data or API

---

### Official Review · Reviewer_ztxV · 2024-11-28

**Novelty:** 3
**Technical Quality:** 3

**Review:**

Summary:
The Seed framework aims to overcome the limitations of existing trajectory generation methods by combining sequence models and diffusion models. It seeks to achieve a balance between consistency, regularity, and diversity in synthetic trajectory generation.

Strengths:
Uses Node2vec to encode road network topologies into embeddings that preserve spatial and topological properties of road segments.

Combines sequence models (e.g., Transformer) for movement regularity and consistency with diffusion models.

Limitations:
Node2vec for road segment embeddings and curriculum learning for staged training are well-considered and effectively strengthen the methodology. However, these techniques are adaptations of well-established methods—Node2vec for graph embeddings (Grover & Leskovec, 2016) and curriculum learning for progressive training (Bengio et al., 2009). While their integration is effective within this paper, it somewhat limits the originality of the overall contribution.

“Bengio, Yoshua, et al. "Curriculum learning." Proceedings of the 26th annual international conference on machine learning. 2009.”

“Grover, Aditya, and Jure Leskovec. "node2vec: Scalable feature learning for networks." Proceedings of the 22nd ACM SIGKDD international conference on Knowledge discovery and data mining. 2016.”

Figure 3 illustrates representative trajectories generated by Seed, highlighting consistency and diversity. However, the figure lacks exploration of how the model performs in noisy conditions.

Table 1 highlights Seed's strong performance over traditional baselines, but it does not include comparisons with recent state-of-the-art methods that utilize Graph Neural Networks (GNNs) or Transformer-based architectures. For instance, the framework Adaptive Trajectory Prediction via Transferable GNN employs a GNN-based approach to model agent interactions and achieves adaptive, transferable trajectory predictions. Such methods provide strong baselines for trajectory-related tasks, and their inclusion could further contextualize Seed's performance.

“Xu, Yi, et al. "Adaptive trajectory prediction via transferable gnn." Proceedings of the IEEE/CVF conference on computer vision and pattern recognition. 2022.”

While the paper compares Seed to traditional baselines (e.g., SeqGAN, TrajVAE), it excludes more recent state-of-the-art methods, such as GNN-based or Transformer-based trajectory models. For example, Adaptive Trajectory Prediction via Transferable GNN (Xu et al., CVPR 2022) models agent interactions effectively and could serve as a strong baseline for trajectory-related tasks.

The experimental evaluation is limited to structured urban datasets, namely Porto, Shenzhen, and Chengdu, which are characterized by well-defined and dense road networks. While these datasets effectively showcase the model's performance in urban scenarios, they do not provide insight into its generalizability to rural or irregular road networks, which often feature sparse connectivity.

**Questions:**

1. How does the model handle noisy trajectory data? (Figure 3)
2. Why were advanced baselines like GNN-based methods excluded from the comparison? (Table 1)
3. How does curriculum learning scale with larger datasets or more complex road networks? (Figure 2)
4. What are the hardware requirements for deploying Seed in real-world systems? (Table 6)
5. Can Seed simultaneously optimize diversity and regularity metrics? (Figure 8)
6. Table 6 presents the dataset statistics, but how does Seed scale to larger road networks and perform in real-time applications?

**Reviewer Confidence:**

3: The reviewer is confident but not certain that the evaluation is correct

**Scope:**

3: The work is somewhat relevant to the Web and to the track, and is of narrow interest to a sub-community

---

### Official Review · Reviewer_mipZ · 2024-12-01

**Novelty:** 4
**Technical Quality:** 5

**Review:**

This paper introduces an approach that combines sequence models with diffusion-based models to enhance the consistency and diversity in the Road Trajectory Generation task. The paper is well-structured and clearly written, with a thorough explanation of the proposed method. Experimental results demonstrate competitive performance.

**Questions:**

However, a potential concern is that the focus on Road Trajectory Generation may not align closely with the core topics of the WWW conference. Submitting this work to conferences such as AAAI, IJCAI, or NeurIPS, which may be more aligned with similar studies, could be a better fit.

**Reviewer Confidence:**

3: The reviewer is confident but not certain that the evaluation is correct

**Scope:**

2: The connection to the Web is incidental, e.g., use of Web data or API

---

### Official Review · Reviewer_7j7s · 2024-12-02

**Novelty:** 4
**Technical Quality:** 4

**Review:**

The paper proposes a novel approach to road trajectory generation by combining sequence and diffusion models' strengths. The authors identify that existing methods either excel in consistency and regularity but lack diversity, or achieve high diversity at the expense of consistency and regularity. The paper introduces Seed, a method that adopts a conditional diffusion structure to bridge sequence and diffusion models for trajectory generation to address these limitations. Seed uses a Transformer to model the movement of each trajectory along road segments and conditions a diffusion model on the Transformer's output to recover the next road segment from random noise. This approach aims to capture sequential movement patterns for regularity and consistency while introducing diversity through the diffusion model's recovery from noise. The paper also presents a training strategy that includes preparing model inputs using Node2vec embeddings for road segments, supervising learning with the task of trajectory reconstruction, and utilizing curriculum learning to accelerate convergence. Seed is evaluated on three datasets and compared with eight state-of-the-art trajectory generation methods, showing significant improvements in consistency and matching the best-performing baselines in regularity and diversity.

However, some drawbacks in the paper are as follows: 1) Seed’s key idea of combination with Transformer and diffusion model has been proposed in the previous research in NLP[1] and CV[2,3] domains, which limits the novelty of the key contribution. 2) And the domain-specific designs such as transition frequency-based pre-trained feature as input [4] are also used in the transportation domains, which limits the creativity of the model designing. 3) Moreover, trajectories are strictly constrained by the road network structure (i.e., there are only several reasonable path Candidates for a specific origin and destination so that the diversity and consistency are limited), the significance of evaluating regularity, consistency, and diversity makes less sense than scenarios like text sentences in NLP and images in CV.

So the reviewer thinks that the paper mainly evaluates the transfer-ability of the architecture of extracting the Transformer and diffusion model for trajectory generation tasks in transportation domains. The key of Seed is to propose a novel sequence and diffusion model for trajectory generation, in which the useful module designs include curriculum training strategy, auxiliary denoising task, .etc. Meanwhile, the manuscript contains solid experiments and detailed analysis.

[1] Li, Xiang, et al. "Diffusion-lm improves controllable text generation." Advances in Neural Information Processing Systems 35 (2022): 4328-4343.

[2] Chen, Shoufa, et al. "Diffusiondet: Diffusion model for object detection." Proceedings of the IEEE/CVF international conference on computer vision. 2023.

[3] Hertz, Amir, et al. "Prompt-to-Prompt Image Editing with Cross-Attention Control." The Eleventh International Conference on Learning Representations.

[4] Mao, Zhenyu, et al. "Jointly contrastive representation learning on road network and trajectory." Proceedings of the 31st ACM International Conference on Information & Knowledge Management. 2022.

**Questions:**

1) The Chengdu and Porto datasets are widely used in the research. But the URL of the Shenzhen Dataset the authors provided in the manuscript is not available, is this the correct URL (https://people.cs.rutgers.edu/~dz220/data.html)? If yes, the reviewer wonders about the data pre-processing details such as the departure time range of the dataset the authors utilized, and the way in which timestamped GPS points are identified and constructed into the ordered trajectories.

2) As the time complexity of the standard Transformer approximately is O(d\*n\^2), and the diffusion model is O(nT), the time complexity of Seed may be O((d\*n\^2)\*[T/S]). The reviewer wonders the efficiency of Seed compared with other baselines.

**Reviewer Confidence:**

4: The reviewer is certain that the evaluation is correct and very familiar with the relevant literature

**Scope:**

3: The work is somewhat relevant to the Web and to the track, and is of narrow interest to a sub-community

---

### Official Review · Reviewer_EBGb · 2024-12-02

**Novelty:** 3
**Technical Quality:** 4

**Review:**

This paper introduces an approach for trajectory generation by combining DiffTraj [43] and SVAE [12] (citations consistent with the paper). The proposed combination leverages the strengths of both methods: DiffTraj excels in generating diverse trajectories, while SVAE produces trajectories with strong consistency and regularity. The integration is achieved by using the latent representation learned from SVAE as a condition for the DiffTraj model. Extensive experiments demonstrate promising results, showcasing the effectiveness of this combination.
While the experiments highlight the strong performance of this approach, the innovation appears to be limited, as the work primarily focuses on combining existing methods rather than introducing a fundamentally new technique. Below are the identified pros and cons of this work:

Pros
1.	The combination of DiffTraj and SVAE successfully balances trajectory diversity, regularity, and consistency, addressing key challenges in trajectory generation.
2.	The introduction of curriculum learning is a noteworthy contribution, as it accelerates training and enhances model performance.
3.	Extensive experimental evaluation demonstrates superior performance compared to various baselines, adding credibility to the proposed approach.

Cons
1.	The level of innovation is limited, as the proposed solution is primarily a combination of two existing methods rather than a novel technique.
2.	While the integration of DiffTraj and SVAE is effective, the authors do not provide a detailed justification for their specific design choices or discuss the unique challenges encountered in combining these approaches, which diminishes the perceived novelty

**Questions:**

1.	The paper effectively combines elements from DiffTraj and SVAE. Could the authors clarify specific challenges faced in integrating these approaches and how they were addressed?
2.	The paper highlights the use of Node2Vec embeddings. Have the authors experimented with alternative embedding strategies, such as graph neural networks or Chen et.al. 2021, and how do they compare?
Chen, Y., Li, X., Cong, G., Bao, Z., Long, C., Liu, Y., ... & Ellison, R. (2021, October). Robust road network representation learning: When traffic patterns meet traveling semantics. In Proceedings of the 30th ACM International Conference on Information & Knowledge Management (pp. 211-220).

**Reviewer Confidence:**

4: The reviewer is certain that the evaluation is correct and very familiar with the relevant literature

**Scope:**

4: The work is relevant to the Web and to the track, and is of broad interest to the community